# *Aquibium pacificus* sp. nov., a Novel Mixotrophic Bacterium from Bathypelagic Seawater in the Western Pacific Ocean

**DOI:** 10.3390/microorganisms12081584

**Published:** 2024-08-04

**Authors:** Fan Jiang, Xun Hao, Ding Li, Xuying Zhu, Jiamei Huang, Qiliang Lai, Jianning Wang, Liping Wang, Zongze Shao

**Affiliations:** 1Key Laboratory of Marine Genetic Resources, Third Institute of Oceanography, Ministry of Natural Resources of PR China, Xiamen 361102, China; 2State Key Laboratory Breeding Base of Marine Genetic Resources, Xiamen 361102, China; 3College of Ocean and Earth Sciences, Xiamen University, Xiamen 361102, China

**Keywords:** *Aquibium pacificus* LZ166^T^, taxonomy, mixotrophy, carbon fixation, sulfur oxidation

## Abstract

A novel Gram-stain-negative, facultatively anaerobic, and mixotrophic bacterium, designated as strain LZ166^T^, was isolated from the bathypelagic seawater in the western Pacific Ocean. The cells were short rod-shaped, oxidase- and catalase-positive, and motile by means of lateral flagella. The growth of strain LZ166^T^ was observed at 10–45 °C (optimum 34–37 °C), at pH 5–10 (optimum 6–8), and in the presence of 0–5% NaCl (optimum 1–3%). A phylogenetic analysis based on the 16S rRNA gene showed that strain LZ166^T^ shared the highest similarity (98.58%) with *Aquibium oceanicum* B7^T^ and formed a distinct branch within the *Aquibium* genus. The genomic characterization, including average nucleotide identity (ANI, 90.73–76.79%), average amino identity (AAI, 88.50–79.03%), and digital DNA–DNA hybridization (dDDH, 36.1–22.2%) values between LZ166^T^ and other species within the *Aquibium* genus, further substantiated its novelty. The genome of strain LZ166^T^ was 6,119,659 bp in size with a 64.7 mol% DNA G+C content. The predominant fatty acid was summed feature 8 (C_18:1_*ω*7*c* and/or C_18:1_*ω*6*c*). The major polar lipids identified were diphosphatidylglycerol (DPG), phosphatidylethanolamine (PE), glycolipid (GL), and phosphatidylglycerol (PG), with ubiquinone-10 (Q-10) as the predominant respiratory quinone. The genomic annotation indicated the presence of genes for a diverse metabolic profile, including pathways for carbon fixation via the Calvin–Benson–Bassham cycle and inorganic sulfur oxidation. Based on the polyphasic taxonomic results, strain LZ166^T^ represented a novel species of the genus *Aquibium*, for which the name *Aquibium pacificus* sp. nov. is proposed, with the type strain LZ166^T^ (=MCCC M28807^T^ = KACC 23148^T^ = KCTC 82889^T^).

## 1. Introduction

The genus *Aquibium*, derived from its isolation from aquatic environments, was first proposed by Kim et al. and classified within the family Phyllobacteriaceae [1]. At the time of writing, only three species within the genus *Aquibium* have been validly published according to the List of Prokaryotic names with Standing in Nomenclature (LPSN, http://lpsn.dsmz.de/search?word=Aquibium accessed on 20 June 2024) [2]. The type species is *Aquibium microcysteis*, with the other two species, *Aquibium carbonis* and *Aquibium oceanicum*, having been reclassified from the genus *Mesorhizobium* [3]. While most of the species of *Mesorhizobium* were isolated from the rhizospheres of leguminous plants, the *Aquibium* species originated from aquatic environments such as coal bed water [4], deep-sea water [5], and cultures of *Microcystis aeruginos* [1]. The genomic analysis suggested that *Aquibium* species might be ancestral to those in *Mesorhizobium*, with some *Mesorhizobium* members acquiring nitrogen-fixing genes over the course of evolution [6]. Species within *Aquibium* consistently exhibit characteristics such as being Gram-straining-negative, aerobic, and catalase- and oxidase-positive. The genomic DNA G+C content, predominant fatty acid, and respiratory quinone for these species are 65.1–67.9 mol%, summed feature 8 (C_18:1_*ω*7*c* and/or C_18:1_*ω*6*c*), and ubiquinone-10 (Q-10), respectively [1].

Marine microorganisms play a crucial role in global carbon cycling and ecological interactions. Mixotrophy, a key trophic strategy involving the concurrent use of both autotrophic and heterotrophic nutrition, significantly influences these cycles [7]. This metabolic flexibility, as observed in ubiquitous species like *Prochlorococcus* [8], SAR324 [9], and *Arcobacteraceae* [10], enables adaptation to the fluctuating marine environment [11]. While some progress has been achieved, gaps still exist in the aspect of the mixotrophic microorganisms, their metabolic mechanisms, and their ecological roles.

In this study, a novel strain, designated as LZ166^T^, was isolated from bathypelagic seawater in the western Pacific Ocean. A comprehensive investigation of the morphology, physiology, chemotaxonomy, and phylogeny was conducted to elucidate its taxonomy status and consequently denominate it as *Aquibium pacificus* sp. nov. Furthermore, genomic analyses were performed to delineate the mixotrophic lifestyle supported by the flexible metabolic functions of carbon, nitrogen, and sulfur in this novel strain.

## 2. Materials and Methods

### 2.1. Isolation and Culture

Strain LZ166^T^ was isolated from the bathypelagic seawater of the western Pacific Ocean at station CTD-11 (1000 m depth, 16°13′12″ N, 130°22′12″ E) during the DY60 cruise in January 2021. Seawater samples were collected and filtered through a 0.2 μm polycarbonate (PC) membrane (Merck, Rahway, NJ, USA) on board. The membrane was then immediately transferred to Axygen screw cap tubes containing 30% (*v*/*v*) glycerol, and subsequently stored at −80 °C for preservation prior to laboratory analysis. Once in the laboratory, microbes attached to the membrane were washed off and serially diluted with sterile artificial seawater and spread onto Marine Agar (MA, BD Difco, Sparks, MD, USA) plates. Individual colonies were isolated and transferred to fresh MA plates to obtain pure cultures. Strain LZ166^T^ was successfully isolated and cultivated following these procedures. For long-term preservation, cultures were mixed with 30% glycerol and stored at −80 °C.

Closely related type strains, including *A. microcysteis* NIBR3^T^ and *A. oceanicum* B7^T^, were purchased from the Korean Agricultural Culture Collection (KACC) and the Marine Culture Collection of China (MCCC), respectively. These type strains were used for comparison of phenotypic, physiological, and chemotaxonomic characteristics with the isolated strain LZ166^T^ in this study.

Additionally, strain LZ166^T^ has been deposited in multiple culture collections, including the MCCC (accession number = MCCC M28807^T^), the KACC (accession number = KACC 23148^T^), and the Korean Culture Type Collection (KCTC, accession number = KCTC 82889^T^).

### 2.2. Morphology and Physiology

The cell size, morphology, and flagella pattern during the mid-exponential growth phase were examined using a transmission electron microscope (model HT7800, Hitachi, Tokyo, Japan) after cells being negatively stained with uranyl acetate. The Gram-staining experiment was carried out using a commercial Gram-staining kit (Qingdao Hope Bio-Technology Co., Ltd., Qingdao, China) following the manufacturer’s protocol. The motility of strain LZ166^T^ was assessed both microscopically and by the semi-solid agar puncture method in MA medium containing 0.5% (*w*/*v*) agar. The growth of strain LZ166^T^ was examined in Marine Broth 2216 (MB, BD Difco) medium under a range of temperatures (4, 10, 15, 25, 28, 30, 34, 37, 41, 45, 55, and 60 °C), NaCl concentrations (0, 0.5, 1, 2, 3, 4, 5, 6, 7, 8, 9, 10, 12, 15, 18, and 20%, *w*/*v*), and pH conditions (3.0, 4.0, 5.0, 6.0, 7.0, 8.0, 9.0, 10.0, 11.0, and 12.0), respectively. For NaCl concentrations, 0–2% and 3–20% mediums were prepared based on 2216E and MB medium, respectively. The pH of MB medium was adjusted using various buffers to cover the desired pH range: 10 mM acetate/acetic acid buffer for pH 3.0–5.5, MES buffer for pH 5.0–6.5, PIPES buffer for pH 6.0–7.0, HEPES buffer for pH 7.0–8.0, and Tris and CAPSO buffer for pH values greater than 8.0 [12]. The anaerobic growth was determined in an anaerobic chamber with pure N_2_ using MB medium. Growth on nutrient agar (NA), R2A agar (BD Difco), and LB agar plates was also tested at 28 °C.

Catalase activity was assessed by the formation of oxygen bubbles when the cells were exposed to a 3% (*w*/*v*) H_2_O_2_ solution [1]. Oxidase activity was determined by the oxidation of 1% (*w*/*v*) N,N,N,N-tetramethyl-1,4-phenylenediamine (BioMérieux, Lyon, France). The hydrolysis of skimmed milk, starch, cellulose, Tween 20, 40, 60, and 80 was examined on MA plates supplemented with the corresponding substrates. Additional biochemical properties were identified using API ZYM [13] and 20NE strips (BioMérieux, Lyon, France) following the manufacturer’s protocol. Utilization of various carbon sources by strain LZ166^T^ was assessed with the Biolog GEN III Microplates (Biolog, Hayward, CA, USA) following the manufacturer’s instructions.

### 2.3. Chemotaxonomy

To analyze the cellular fatty acid and polar lipids, strain LZ166^T^ and closely related species were cultivated in MA medium at 28 °C for 72 h. Fatty acids were extracted, saponified, methylated, and then determined using gas chromatography (Agilent Technologies 6850, Santa Clara, CA, USA) as described by Li et al. [12]. The composition of isoprenoid quinones from freeze-dried strains was extracted using chloroform/methanol (2:1, *v*:*v*), separated by thin-layer chromatography (TLC) on silica gel GF254 plates (10 × 20 cm, Qingdao Haiyang Chemical Co., Ltd., Qingdao, China), and redissolved in methanol. Redissolved isoprenoid quinones were then analyzed using high-performance liquid chromatography (HPLC, Waters 2695, Milford, MA, USA) [14]. Polar lipids were extracted and separated using two-dimensional TLC on silica gel 60 F254 plates (10 × 10 cm, Merck) with a first phase of chloroform/methanol/water (65:25:4, *v*:*v*:*v*) and a second phase of chloroform/acetic acid/methanol/water (80:15:12:5, *v*:*v*:*v*:*v*). The polar lipids were visualized using molybdatophosphoric acid, ninhydrin, molybdenum blue reagent, and 1-naphthol-sulphuric acid and identified based on their relative positions under different chromogenic agents [14].

### 2.4. 16S rRNA Gene Phylogeny

The genomic DNA of strain LZ166^T^ was extracted using a bacterial genomic DNA extraction kit (SBS Genetech Co., Ltd., Shanghai, China) according to the manufacturer’s instructions. The purified genomic DNA was quantified by NanoDrop 2000 spectrophotometer (Thermo Scientific, Waltham, MA, USA). The complete 16S rRNA gene was amplified using universal primers 27F (5′-AGAGTTTGATCMTGGCTCAG-3′) and 1492R (5′-GGTTACCTTGTTACGACTT-3′). Further, 16S rRNA genes were sequenced via the Sanger method (Applied BiosystemsTM 3730XL, Waltham, MA, USA) by Sangon Biotech (Shanghai) Co., Ltd., Shanghai, China. The 16S rRNA gene sequence of LZ166^T^ was subjected to BLAST analysis on the EzBioCloud platform (https://www.ezbiocloud.net/identify accessed on 20 June 2024) [15]. Fifty 16S rRNA gene sequences of reference species were downloaded from the NCBI database. All 16S rRNA gene sequences were trimmed, aligned, and then used to construct phylogenetic trees using MEGA software (v 7.0.21) [16]. The trees were generated with 1000 bootstrap iterations and clustering using maximum likelihood (ML) [17], neighbor joining (NJ) [18], and minimum evolution (ME) [19] methods. Evolutionary distances were calculated using Kimura 2-parameter model [20]. The phylogenetic trees were visualized and decorated using ChiPlot (https://www.chiplot.online/ accessed on 1 July 2024) [21]. The complete 16S rRNA gene sequence was deposited in the GenBank database under accession number PP812687.

### 2.5. Genome Sequencing, Phylogenomic Analysis, and Comparative Analysis

The genome DNA of strain LZ166^T^ was sequenced using the pair-end method on the Illumina HiSeq X Ten sequencing platform by Shanghai Majorbio Bio-Pharm Technology Co., Ltd. (Shanghai, China). The high-quality reads (Q30) were assembled using SPAdes software (v 3.13.0) [22]. Gene prediction was performed using Prokka (v 1.13) [23]. In addition, rRNA and tRNA genes were identified using RNAmmer (v 1.2) [24] and ARAGORN (v 1.2.41) [25]. Genome sequences of reference species were obtained from NCBI. The G+C content of the chromosomal DNA was determined directly from the genome sequence. A phylogenomic tree based on 13 genome sequences was constructed using PhyloPhlAn 3.0 (v 3.0.58) [26]. Average nucleotide identity (ANI) and average amino identity (AAI) among strain LZ166^T^ and reference strains were calculated using PYANI (v 0.2.9) [27] and CompareM (v 0.0.32, https://github.com/donovan-h-parks/CompareM accessed on 20 June 2024), respectively. Digital DNA–DNA hybridization (dDDH) estimated values were calculated using the GGDC 3.0 on TYGS (https://ggdc.dsmz.de/ggdc.php accessed on 20 June 2024) [28]. The genomic overview was presented based on rapid annotation using the RAST server (https://rast.nmpdr.org/rast.cgi accessed on 20 June 2024) [29]. The genome of strain LZ166^T^ was annotated against clusters of orthologous groups (COG) and carbohydrate-active enzyme (CAZy) databases using Eggnog-mapper (v 2.0) [30]. The carbon, nitrogen, and sulfur metabolic pathways were annotated using METABOLIC (v 4.0) [31] and KofamKOALA [32] against the KEGG database. The draft genome sequence of strain LZ166^T^ was deposited in GenBank under accession number JBDPGJ000000000.

## 3. Results and Discussion

### 3.1. Morphology and Physiology

Strain LZ166^T^ is Gram-stain-negative, facultatively anaerobic, and catalase- and oxidase-positive. The cells are short rod-shaped (1.3–1.7 μm long; 1.0–1.1 μm wide), non-spore-forming, and motile by a lateral flagella (Appendix A). After incubation for 3 days on an MA plate at 28 °C, circular (1–2 mm in diameter), convex, smooth, creamy-white, and non-transparent colonies were observed. Growth was observed at temperatures ranging from 10 to 45 °C (optimum 34–37 °C), in NaCl concentrations of 0–5% (*w*/*v*) (optimum 1–3%), and at a pH range of 5–10 (optimum pH 6–8). The physiological and biochemical differences among strain LZ166^T^, *A. microcysteis* NIBR3^T^, and *A. oceanicum* B7^T^ are detailed in Table 1. The strain was positive for skimmed milk hydrolysis but negative for cellulose, starch, Tween 20, 40, 60, and 80 hydrolysis. Additionally, the R2A and LB plates supported the heterotrophic growth of LZ166^T^ but the NA plate did not. Anaerobic growth was observed in the MB medium. While most of the API ZYM results were consistent between LZ166^T^ and the reference species, differences were noted in the activity of alkaline phosphatase and the utilization of D-mannitol, potassium gluconate, and malic acid. Moreover, the Biolog test suggested that LZ166^T^ was capable of utilizing a range of organic carbon sources, including dextrin, D-cellobiose, gentiobiose, D-turanose, α-D-glucose, and D-fucose, for heterotrophic growth (Appendix A).

### 3.2. Chemotaxonomy

The respiratory quinones of strain LZ166^T^ were determined as Q-10 (100%). The predominant respiratory quinones are consistent with the ubiquinone systems characteristic of the genus *Aquibium* [1]. The polar lipids of LZ166^T^ were composed of diphosphatidylglycerol (DPG), phosphatidylethanolamine (PE), glycolipid (GL), and phosphatidylglycerol (PG) (Appendix A). DPG, PG, and PE were also detected in *A. microcysteis* NIBR3^T^, *A. carbonis* B2.3^T^, and *A. oceanicum* B7^T^ [1], which supported the affiliation of strain LZ166^T^ with *Aquibium*. However, the presence of GL can be used to differentiate the novel strain from the reference species as it is unique to strain LZ166^T^. The major cellular fatty acids (>5%) of strain LZ166^T^ are summed feature 8 (C_18:1_*ω*7*c* and/or C_18:1_*ω*6*c*, 39.3%), iso-C_17:0_ (13.3%), 11-methyl C_18:1_*ω*7*c* (12.0%), C_19:0_ cyclo *ω*8*c* (9.3%), and C_16:0_ (5.5%) (Table 2 and Appendix A). Although the fatty acid profiles of strain LZ166^T^ are similar to those of the reference strains in *Aquibium*, their proportions differ from each other [1].

### 3.3. 16S rRNA Gene Phylogeny

The 16S rRNA gene sequence analysis indicated that LZ166^T^ exhibited the highest sequence similarity of 98.58% with *A. oceanicum* B7^T^, followed by *A. carbonis* B2.7^T^ (97.94%) and *A. mycrocysteis* NIBR3^T^ (96.71%), which were lower than the cut-off value of 98.65% [33]. A detailed ML tree (Appendix A) and a concise ML tree (Figure 1) based on the 16S rRNA gene sequences were constructed and both of them clearly revealed that strain LZ166^T^ formed a separate lineage within the genus *Aquibium*. This topology was further confirmed by the ME (Appendix A) and NJ methods (Appendix A). The collective findings from the 16S rRNA gene phylogeny suggest that strain LZ166^T^ could represent a potential novel species within *Aquibium*.

### 3.4. Genomic Features

A total of 10,595,752 raw reads were produced with 258X sequencing depth. The assembled genome size of strain LZ166^T^ is 6,119,659 bp with a chromosomal DNA G+C content of 64.7 mol%, which is similar to the other species within the genus *Aquibium*, which ranges from 65.1–67.9 mol% [1]. Moreover, fifty tRNA genes for twenty amino acids as well as one gene each for 5S rRNA, 16S rRNA, and 23S rRNA were also identified in the genome of LZ166^T^. A total of 5513 protein-coding sequences (CDSs) were predicted, with the majority (5220/5513, 94.7%) assigned to a putative function based on the COG categories, while the rest were annotated as hypothetical proteins. Detailed information on the gene classification according to the RAST, COG, and CAZy databases is provided in Appendix A.

The ANI and AAI values between LZ166^T^ and the other closely related species in the genus *Aquibium* are in the ranges of 76.79–90.73% and 79.03–88.50%, respectively (Appendix A). These values are below the 95–96% cut-off thresholds previously proposed for species delineation [34,35,36]. The dDDH estimated values between LZ166^T^ and the other three species, *A. microcysteis* NIBR3^T^, *A. oceanicum* B7^T^, and *A. carbonis* B2.3^T^, were 22.8%, 36.1%, and 22.2%, respectively (Appendix A), all of which are below the dDDH standard cut-off value of 70% [36,37]. Furthermore, a whole-genome phylogenomic tree (Figure 2) was also constructed, which corroborated the phylogenetic relationships derived from the 16S rRNA genes’ phylogeny and genetic relatedness, as depicted in Figure 1. Altogether, these results suggest that strain LZ166^T^ represents a novel species within the genus *Aquibium*.

According to the RAST annotation, 1398 genes were detected in the genome of strain LZ166^T^ and could be assigned to 279 subsystems belonging to 25 categories. Among the 25 categories, amino acids and derivatives (241) was the most common, followed by carbohydrates (193), protein metabolism (172), and cofactors, vitamins, prosthetic groups, and pigments (107). Most of the genes in the amino acids and derivatives category are connected to branched-chain amino acids. Most of the genes in the carbohydrates category were relevant to the TCA cycle belonging to the central carbohydrate metabolism subcategory (Appendix A). The COG analysis suggested that, except those with unknown functions (S, 22.45%), amino acid transport and metabolism (E, 12.53%) was the most abundant category, followed by carbohydrate transport and metabolism (G, 7.91%), transcription (K, 7.78%), energy production and conversion (C, 7.45%), and lipid transport and metabolism (I, 6.84%) (Appendix A). Additionally, a total of 71 genes in the genome of strain LZ166^T^ matched the CAZy families. Glycosyl transferases (GTs) was the largest CAZy family with thirty-eight genes, followed by glycoside hydrolases (GHs, twenty-nine genes), carbohydrate-binding modules (CBMs, six genes), carbohydrate esterases (CEs, three genes), and polysaccharide lyases (PLs, one gene). However, auxiliary activities (AAs) were not detected (Appendix A).

### 3.5. Genomic Functional Analysis

#### 3.5.1. Carbon Metabolism

The draft genome sequence of strain LZ166^T^ was utilized to infer its metabolic profiles, which was subsequently compared with those of three other *Aquibium* species (Figure 2). LZ166^T^ is capable of obtaining carbon from both inorganic and organic sources. A distinctive feature of the LZ166^T^ genome is its potential for carbon dioxide fixation via the Calvin–Benson–Bassham (CBB) cycle. The genome contains a complete set of genes necessary for the CBB cycle, including those encoding key enzymes such as ribulose-1,5-bisphosphate carboxylase (Rubisco) and phosphoribulokinase (PRK) [38]. The Rubisco enzyme catalyzes the carboxylation of ribulose-1,5-bisphosphate, resulting in the formation of 3-phosphoglycerate [39]. To date, four forms of Rubisco have been identified, with form I being the most prevalent [40]. The genome of LZ166^T^ encodes form I RubisCO (RcbL and RcbS), which shows the highest similarity (88.8% and 84.7%, respectively) to the enzymes from *Mesorhizobium mediterraneum*. Although the *rcbL* and *rcbS* genes were predicted in the genome of LZ166^T^, no complete carbon fixation pathways have been identified in the genomes of the other *Aquibium* species. This indicates that the *rcbL*, *rcbS*, and *prk* genes may serve as genetic markers to distinguish strain LZ166^T^ from the other members of the genus. The presence of the CBB pathway hints that LZ166^T^ may be capable of autotrophic metabolism. Additionally, the genome of strain LZ166^T^ also encodes the genes involved in the Embden–Meyerhof pathway (EMP), hexose monophosphate pathway (HMP), Entner–Doudoroff pathway (ED), and tricarboxylic acid cycle (TCA), indicating a heterotrophic lifestyle that relies on organic matter for carbon and energy [41]. Overall, the presence of both carbon fixation and organic matter breakdown genes in the genome of LZ166^T^ suggests a mixotrophic strategy, potentially allowing it to adapt to the dynamic and variable conditions of the marine environment [11].

The genome of strain LZ166^T^, in common with those of the other *Aquibium* species, includes all the key genes that encode the aerobic carbon monoxide dehydrogenase (CoxSLM). While the other *Aquibium* species only contain the form II *cox* gene, LZ166^T^ distinctively harbors both the form I and form II *cox* genes (Figure 2 and Appendix A). Moreover, at least three copies of *cox* gene clusters could be found in the genome of LZ166^T^. This dual presence of *cox* genes indicates that LZ166^T^ might have the capacity to oxidize carbon monoxide (CO). The co-existence of the CBB pathway for carbon fixation and the CO oxidation pathway in the genome of LZ166^T^ suggests that this strain could utilize the energy derived from CO oxidation. This capability might, in turn, facilitate its autotrophic growth [42]. It seems not surprising as some CO oxidizers have been confirmed to fix CO_2_ through the CBB cycle by using the energy obtained from CO oxidation [42].

#### 3.5.2. Sulfur Metabolism

Sulfur metabolism is crucial for bacteria, providing not only the essential sulfur element but also a source of energy. The genome of strain LZ166^T^ encodes the complete set of genes for the SOX pathway (*soxABCDXYZ*), which is responsible for oxidizing thiosulfate to sulfate [43]. In addition to the SOX system, the LZ166^T^ genome also includes genes encoding flavocytochrome c-sulfide dehydrogenase (*fcc*) and sulfide:quinone oxidoreductase (*sqr*) genes, indicating that this strain potentially possesses the capacity to oxidize sulfide to elemental sulfur [43]. The presence of the SOX system, *fcc*, and *sqr* in the other *Aquibium* species suggests that sulfur oxidation is probably a common feature of this genus. However, an exception is *A. oceanicum* B7^T^, which lacks the *fcc* gene despite its phylogenetic proximity to strain LZ166^T^. Furthermore, the *Aquibium* species, including LZ166^T^, encodes the sulfite dehydrogenases (*soe*) gene, which plays a role in sulfite oxidation [43,44]. Strain LZ166^T^ and the other *Aquibium* species possessed a complete assimilatory sulfate reduction pathway (*cysND*, *cysC*, *cysH*, and *cysJI*), indicating that they can reduce sulfate into sulfide, even to L-cysteine (*cysK*) [45]. The presence of multiple sulfur oxidation pathways in the genome of LZ166^T^ underscores its capacity to oxidize sulfide, sulfite, or thiosulfate, thereby conserving energy and potentially supporting chemoautotrophic growth.

#### 3.5.3. Nitrogen Metabolism

The analysis of nitrogen metabolism using KEGG annotation revealed the absence of nitrogenase (*nifH*) genes in strain LZ166^T^ as well as in the other *Aquibium* species, a distinguishing feature between the genera *Aquibium* and *Mesorhizobium* [1]. Strain LZ166^T^ was found to harbor an incomplete denitrification pathway, characterized by the presence of key genes of *nirK* and *nosZ*, which encode nitrite reductase [46] and nitrous oxide reductase [47], respectively. This genetic profile implies the potential for LZ166^T^ to utilize nitrite or nitrous oxide as electron acceptors. However, neither the nitric oxide reductase genes (*norB*) [48] nor the dissimilatory nitrate reductase genes (*napA* and *narG*) [46] and the nitrite reductase genes (*nrfAH* and *nirBD*) [49] could be identified in the genome. Additionally, the assimilatory nitrate reduction pathway was annotated in the genome of LZ166^T^, which contained the *nasDE* genes, responsible for the conversion of nitrite to ammonia [50]. However, the assimilatory nitrate reductase gene, *nasA* [51], is absent in the LZ166^T^ genome. The genome of strain LZ166^T^ encodes the glutamine synthase (*glnA*) and glutamate synthase (*gltB*) genes, enabling the assimilation of ammonium [51]. Furthermore, the presence of the urease (*ure*) and glutamate dehydrogenase (*gdh*) genes provided genomic evidence for strain LZ166^T^ to transform nitrogen from organic to inorganic forms [46,51].

A comprehensive overview of the metabolic pathways involved in the carbon, nitrogen, and sulfur cycles for strain LZ166^T^ (Figure 3) and the reference strains of the genus *Aquibium* is presented in Figure 2. Consequently, the genomic functional analysis underscores the metabolic versatility of strain LZ166^T^.

### 3.6. Description of Aquibium pacificus sp. nov.

*Aquibium pacifica* (pa.ci’ fi.cus. L. gen. adj. Pacificus, pacific, pertaining to the Pacific Ocean).

The cells are short rod-shaped (1.3–1.7 μm long; 1.0–1.1 μm wide) with a lateral flagella, facultatively anaerobic, motile, and Gram-negative. The colonies are 1–2 mm in diameter, convex, smooth, creamy-white, and non-transparent after incubation on an MA plate for 3 days. The strains can grow in the R2A medium and LB medium but not in the NA medium. Growth can be observed at 10–45 °C (optimum 34–37 °C), 0–5% (*w*/*v*) NaCl (optimum 1–3%), and pH 5–10 (optimum pH 6–8). Catalase, oxidase, esterase (C4), esterase lipase (C8), leucine aminopeptidase, trypsin, and α-glucosidase are positive. Alkaline phosphatase, lipase (C14), valine aminopeptidase, cystine aminopeptidase, α-chymotryp, acid phosphatase, and naphtol-AS-B1-phosphoamidase are weakly positive. α-galactosidase, β-galactosidase, α-glucosidase, β-glucosidase, N-acetyl-β-glucosaminidase, α-mannosidase, and α-fucosidase are negative. The strain is able to hydrolyze or assimilate urea, D-glucose, L-arabinose, and D-mannitol and is weakly positive for aesculin hydrolysis. It is unable to reduce nitrate to nitrite, produce indole, ferment D-glucose, hydrolyze arginine or gelatin, or assimilate D-mannose, D-maltose, potassium gluconate, capric acid, adipic acid, malic acid, trisodium citrate, or phenylacetic acid. Skimmed milk utilization is observed. The major fatty acids (>5%) are summed feature 8 (C_18:1_*ω*7*c* and/or C_18:1_*ω*6*c*), iso-C_17:0_ and 11-methyl C_18:1_*ω*7*c*, C_19:0_ cyclo *ω*8*c*, and C_16:0_. DPG, PE, GL, and PG are the major polar lipids. The predominant respiratory quinone is Q-10.

The type strain designated as LZ166^T^ (=MCCC M28807^T^ = KACC 23148^T^ = KCTC 82889^T^) was isolated from deep seawater in the western Pacific Ocean. The genome size is 6,154,543 bp with 64.7% G+C content. The GenBank accession numbers of the 16S rRNA gene sequence and genome sequence of strain LZ166^T^ are PP812687 and JBDPGJ000000000, respectively.

## 4. Conclusions

The study presents the detailed characterization and taxonomic classification of *Aquibium pacificus* sp. nov., a newly identified mixotrophic bacterium isolated from the western Pacific Ocean’s bathypelagic seawater. The 16S rRNA gene sequence analysis and genomic features, including the ANI, AAI, and dDDH values, distinctly place LZ166^T^ within the *Aquibium* genus, yet as a separate lineage. The physiological and biochemical characteristics of LZ166^T^ were also found to be distinct from the other species within the genus, highlighting its novelty. It hints at a versatile metabolic profile, capable of both autotrophic and heterotrophic growth, with a genome that supports carbon fixation via the CBB cycle and inorganic sulfur oxidation. Notably, LZ166^T^ lacks nitrogenase genes, indicating a divergence from the nitrogen-fixing capabilities observed in other closely related genera. Its metabolic versatility, as evidenced by the presence of genes for carbon fixation and organic matter breakdown, positions it as a mixotrophic organism. The findings of this study extend the knowledge of the *Aquibium* species and their ecological roles in the marine environment, emphasizing the adaptability of LZ166^T^ to the dynamic deep-sea ecosystem.

## Figures and Tables

**Figure 1 microorganisms-12-01584-f001:**
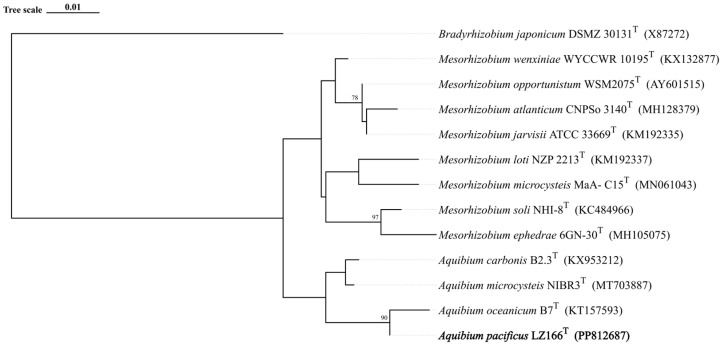
Maximum likelihood phylogenetic tree based on 13 16S rRNA gene sequences showing the positions between strain LZ166^T^ and other closely related phylogenetic neighbors. Bootstrap numbers (>70%) were shown with 1000 calculations. The bold font represents the novel species identified in this study. *Bradyrhizobium japonicum* DSMZ_30131^T^ (X87272) was used as the out group. Bar, 0.01 substitutions per nucleotide position.

**Figure 2 microorganisms-12-01584-f002:**
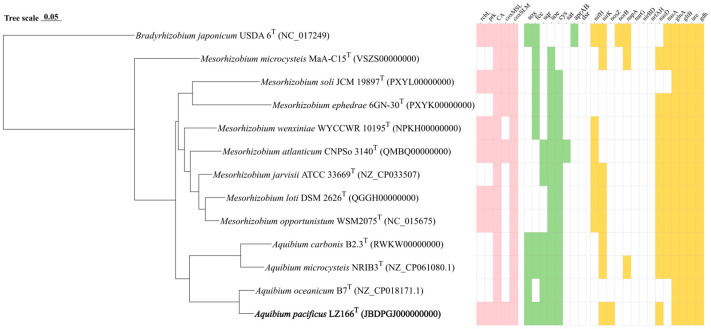
Phylogenomic tree of LZ166^T^ and its functional genes involved in carbon, sulfur, and nitrogen metabolism in comparison with closely related species. The bold font represents the novel species identified in this study. *Bradyrhizobium japonicum* USDA 6^T^ (GFC_000284375.1) was used as the out group. Bar, 0.05 substitutions per nucleotide position. Pink, green, and yellow blocks, presence of corresponding carbon, sulfur, and nitrogen functional genes, respectively. White blocks, absence or partial presence of corresponding functional genes. *rcbL*, ribulose-1,5-bisphosphate carboxylase/oxygenase gene large chain. *prk*, phosphoribulokinase gene. *CA*, carbonic anhydrase gene. *coxMSL*, carbon monoxide dehydrogenase (form I) gene. *coxSLM*, carbon monoxide dehydrogenase (form II) gene. *sox*, thiosulfate oxidation genes cluster (*soxABCDXYZ*). *fcc*, flavocytochrome c-sulfide dehydrogenase gene. *sqr*, sulfide:quinone oxidoreductases gene. *soe*, sulfite dehydrogenase (quinone) gene. *cys*, assimilatory sulfate reduction genes cluster (*cysNDCHJIK*). *sat*, sulfate adenylyltransferase gene. *aprAB*, adenylylsulfate reductase gene. *dsr*, dissimilatory sulfite reductase gene. *nifH*, nitrogenase gene. *nirK*, copper-containing nitrite reductase (denitrification) gene. *nosZ*, nitrous oxide reductase gene. *norB* nitric oxide reductase gene. *napA*, periplasmic dissimilatory nitrate reductase gene. *narG*, membrane-bound dissimilatory nitrate reductase gene. *nirBD*, dissimilatory nitrite reductase (NADH) gene. *nrfAH*, dissimilatory nitrite reductase (cytochrome c-552). *nasD*, assimilatory nitrite reductase gene. *nasA*, assimilatory nitrate reductase. *glnA*, glutamine synthase gene. *gltB*, glutamate synthase gene. *ure*, urease gene. *gdh*, glutamate dehydrogenase gene.

**Figure 3 microorganisms-12-01584-f003:**
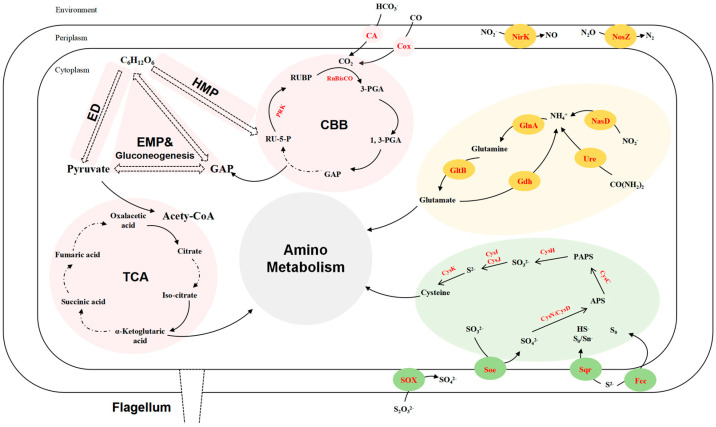
Reconstructed carbon (pink), sulfur (green), and nitrogen (yellow) metabolism of strain LZ166^T^ based on functional genes (corresponding enzymes are in red). ED, Entner–Doudoroff pathway. EMP, Embden–Meyerhof–Parnas pathway. HMP, hexose monophosphate pathway. CBB, Calvin–Benson–Bassham cycle. TCA, tricarboxylic acid cycle. GAP, glyceraldehyde 3–phosphate. RU–5–P, ribulose–5–phosphate. PRK, phosphoribulokinase. RUBP, ribulose–1,5–bisphosphate. RuBisCO, ribulose–1,5–bisphosphate carboxylase/oxygenase. 3–PGA, 3–phosphoglycerate. 1, 3–PGA, 1, 3–diphosphoglycerate. PAPS, phosphoadenosine phosphosulfate. APS, adenosine 5′–phosphosulfate. CA, carbonic anhydrase. Cox, carbon monoxide dehydrogenase. NirK, copper-containing nitrite reductase. NosZ, nitrous oxide reductase. NasD, assimilatory nitrite reductase. GlnA, glutamine synthase. GltB, glutamate synthase. Gdh, glutamate dehydrogenase. Ure, urease. CysK, cysteine synthase. CysI, sulfite reductase (NADPH) hemoprotein beta-component. CysJ, sulfite reductase (NADPH) flavoprotein alpha-component. CysH, PAPS reductase. CysC, adenylylsulfate kinase. CysN, sulfate adenylyltransferase subunit 1. CysD, sulfate adenylyltransferase subunit 2. SOX, thiosulfate oxidation enzymes. Soe sulfite dehydrogenase. Sqr, sulfide:quinone oxidoreductases. Fcc, flavocytochrome c-sulfide dehydrogenase.

**Table 1 microorganisms-12-01584-t001:** Comparison of the phenotypic characteristics of strain LZ166^T^ with reference strains of the genus *Aquibium*.

Characteristic	LZ166^T^	*A. microcysteis* NIBR3^T^	*A. oceanicum* B7^T^
Cell size (μm)	short rod-shaped (1.3–1.7 × 1.0–1.1)	rod-shaped (1.2–2.5 × 0.4–0.6) ^a^	oval-shaped (1.2–2.5 × 0.4–0.6) ^b^
Flagella	+	unknown	− ^b^
Motility	+	+	− ^b^
Temperature range (optimum) for growth (°C)	10–45 (34–37)	23–45 (33–37) ^a^	25–40 (35) ^b^
NaCl range (optimum) for growth (*w*/*v*, %)	0–5% (1–3%)	0–4% (0%) ^a^	0–8% (3%) ^b^
pH range (optimum) for growth	5–10 (6–8)	6–11 (8) ^a^	5.5–9 (7) ^b^
Skimmed milk hydrolysis	+	−	−
Tween 40 hydrolysis	−	−	+
Tween 60 hydrolysis	−	−	+
Alkaline phosphatase	w	+	w
D-mannitol utilization	+	−	+
Potassium gluconate utilization	−	−	+
Malic acid utilization	−	+	+

+, positive. w, weak. −, negative. Data without superscript were obtained in this study. ^a^ Data were taken from Kim et al. 2022 [1]. ^b^ Data were taken from Fu et al. 2017 [5]. All strains were Gram-stain-negative, aerobic, catalase- and oxidase-positive, able to grow on R2A and LB plates but unable to grow on NA plates or hydrolyze starch, cellulose, Tween 20, and 80. In API ZYM tests, all strains were positive for esterase (C4), esterase lipase (C8), leucine aminopeptidase, and trypsin, weak positive for lipase (C14), valineamino peptidase, cystine aminopeptidase, α-chymotryp, acid phosphatase, naphthol-AS-Bl-phosphoamidase, α-glucosidase, negative for α-galactosidase, β-galactosidase, β-glucuronidase, β-glucosidase, N-acetyl-β-glucosaminidase, α-mannosidase, and α-fucosidase. In API 20E test, all strains were able to utilize urea, D-glucose, and L-arabinose but hydrolyze aesculin weakly, unable to produce indole, ferment D-glucose, utilize arginine, gelatin, D-mannose, N-acetyl-glucosamine, D-maltose, capric acid, adipic acid, trisodium citrate, phnylacetic acid, or reduce nitrate to nitrite, and denitrification.

**Table 2 microorganisms-12-01584-t002:** Comparison of major fatty acids (>5%) between strain LZ166^T^ and closely related *Aquibium* species.

Fatty Acid (%)		LZ166^T^	*A. microcysteis* NIBR3^T^	*A. oceanicum* B7^T^
Saturated	C_16:0_	5.5	3.6	5.3
	C_18:0_	4.1	5.8	6.9
Branched	iso-C_17:0_	13.3	4.3	5.5
	C_19:0_ cyclo *ω*8*c*	9.3	ND	5.4
Unsaturated	11-methyl C_18:1_ *ω*7*c*	12.0	11.6	9.6
Summed feature	8*	39.3	58.4	44.3

ND, not detected. Summed feature 8*, C_18:1_ *ω*7*c*, and/or C_18:1_ *ω*6*c*. All data were obtained in this study.

## Data Availability

The 16S rRNA gene sequence and genome sequence of strain LZ166^T^ are available in GenBank under accession numbers PP812687 and JBDPGJ000000000, respectively.

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
