# Peer review of "Aquibium pacificus sp. nov., a Novel Mixotrophic Bacterium from Bathypelagic Seawater in the Western Pacific Ocean"

_microorganisms, 2024, doi:10.3390/microorganisms12081584_

Round 1

Reviewer 1 Report

Comments and Suggestions for Authors

Dear authors,

I carefully read the manuscript and found it significant as new scientific data. However, there are a number of aspects that should be clarified/reworked/presented to make the text easier for readers to understand.

Here are my observations below.

1. Materials and methods, line 80. Please correct "Gram straining" and replace with "Gram staining". Under this aspect, I recommend the authors to go through the manuscript carefully to correct typographical errors.

2. Results and discussions, line 188 "Anaerobic growth was observed in MB medium" seems to contradict the statement from line 178 "aerobic". If the strain grew both in aerobiosis and in anaerobiosis, it means that it is a facultatively anaerobic strain.

3. Results and discussions, lines 182-184. How could the authors explain the following physiological features of the newly isolated strain, taking into account the fact that it is a marine isolate: (i) a very wide range of salinity; (ii) a very wide range of growth temperature, the optimal value being very high and uncharacteristic of marine bacteria. Please bring additional evidence or explanations.

4. Results and Discussions. In my opinion, the figures, especially figures 1 and 2 (lines 215-221, respectively 292-310) should be constructed differently, being too complex, with very small font, impossible to read. At first glance, these figures look like raw data imported from a graphics software.

Data from the article should be a synthesis in which the essential elements can be easily identified and noticed. The raw data that contain a lot of details as they now appear in the article can be moved to Supplementary Material.

Regarding Figure 1 - The phylogenetic tree is represented in relation to many species and is difficult to read. Therefore, I suggest the authors move Figure 1 to Supplementary Data and build a phylogenetic tree based on the comparison between LZ 166, Am microcysteis and A. oceanus. In this way, readers can have a more synthetic and correct picture of the phylogenetic position of this new species.

The same observation for Figure 2 (lines 292-310). The figure is difficult to read and understand. Therefore, I suggest the authors to build three different figures each with the functional genes involved in carbon, sulfur and nitrogen metabolism.

With best regards!

Comments on the Quality of English Language

I would recommend the authors to carefully read the manuscript once more and correct the typographical errors.

Reviewer 2 Report

Comments and Suggestions for Authors

The authors describe the processes carried out to identify a new species of Aquibium which they named A. pacificus sp. nov. Within their studies are biochemical, genetic and genomic characterizations. I consider that they support their analyses with extensive information, however, there are some suggestions broken down below:

The introduction describes the characteristics of the A. pacificus LZ166T strain and the genetically close species/genera. However, the authors do not explain the importance and application of studying these mixotrophic microorganisms. I consider it necessary to explain it in the introduction of any research.

Lines 73 75: Additionally, strain LZ166T has been deposited in multiple culture collections, including the MCCC (accession number = MCCC M28807T), the KACC (accession number = KACC 23148T) and the Korean Culture Type Collection (KCTC, accession number = KCTC 82889T)

My question is: why didn't they deposit the strain in the ATCC?

Line 128: Reference 16S rRNA genes sequences were downloaded from the NCBI database Please indicate how many reference genes you used for this analysis.

Materials and Methods: 2.5 Genome Sequencing, Phylogenomic and Comparative Analysis. Please add the sequencing depth, if you worked with paired-end or single-end reads as well as the total raw reads obtained. Also, please indicate the value of Q that you consider to determine The high-quality reads (Q30?, Q38?, Q40?, etc).

Line 144-145 A phylogenetic tree based on genome sequences was constructed using PhyloPhlAn 3.0 I think you refer to phylogenomics since you indicate having made the tree using complete genomes.

Line 389: I did not find the genome using the accession number JBDPGJ000000000 within the NCBI genebank.

Round 2

Reviewer 1 Report

Comments and Suggestions for Authors

Dear authors,

 Thank you for the time and effort dedicated to correcting the manuscript. I noticed that you made significant changes, improving the quality of the manuscript and making the article publishable and more accessible to readers.

 With best regards!